# Seed Germination and Growth Improvement for Early Maturing Pear Breeding

**DOI:** 10.3390/plants12244120

**Published:** 2023-12-10

**Authors:** Jialiang Kan, Na Yuan, Jing Lin, Hui Li, Qingsong Yang, Zhonghua Wang, Zhijun Shen, Yeqing Ying, Xiaogang Li, Fuliang Cao

**Affiliations:** 1Institute of Pomology, Jiangsu Key Laboratory for Horticultural Crop Genetic Improvement, Jiangsu Academy of Agricultural Sciences, Nanjing 210014, China; 201800701@jaas.ac.cn (J.K.); thefuries@163.com (N.Y.); lj84390224@126.com (J.L.); lihui7904@163.com (H.L.); yng_23@163.com (Q.Y.); wzh925@163.com (Z.W.); shenjaas@aliyun.com (Z.S.); 2State Key Laboratory of Subtropical Silviculture, Zhejiang A&F University, Hangzhou 311300, China; yeqing@zafu.edu.cn; 3Co-Innovation Center for Sustainable Forestry in Southern China, Nanjing Forestry University, Nanjing 210037, China

**Keywords:** *Pyrus pyrifolia*, seed dormancy, cold temperature treatment, germination, seedling growth

## Abstract

Breeding early maturing cultivars is one of the most important objectives in pear breeding. Very early maturing pears provide an excellent parental material for crossing, but the immature embryo and low seed germination of their hybrid progenies often limit the selection and breeding of new early maturing pear cultivars. In this study, we choose a very early maturing pear cultivar ‘Pearl Pear’ as the study object and investigate the effects of cold stratification, the culture medium, and the seed coat on the germination and growth of early maturing pear seeds. Our results show that cold stratification (4 °C) treatment could significantly improve the germination rates of early maturing pear seeds. A total of 100 days of cold-temperature treatment in 4 °C and in vitro germination on White medium increased the germination rate to 84.54%. We also observed that seed coat removal improved the germination of early maturing pear seeds, with middle seed coat removal representing the optimal method, with a high germination rate and low contamination. The results of our study led to the establishment of an improved protocol for the germination of early maturing pear, which will greatly facilitate the breeding of new very early maturing pear cultivars.

## 1. Introduction

Pear (*Pyrus* spp.) trees are one of the most commercially important fruit trees and are widely cultivated in the temperate regions of the world [1]. As a potential source of dietary fiber, vitamins, and bioactive compounds in the human diet, pear fruits have gained great popularity in Europe and Asia [2]. China produces a wide range of pear varieties, with over 2000 preserved varieties. According to statistics from the National Bureau of Statistics of China, in 2022, the pear cultivation area in China was approximately 920,000 hectares, with a yield of 19 million tons. Pear trees were planted on a larger scale in Xinjiang, Hebei, Shandong, and other regions of China [3]. Over the past 20 years, pear plantation areas and production have increased considerably, but the market is dominated by mid- to late-maturing cultivars [4]. These cultivars have similar fruit-maturing periods, a concentrated market time, and a high sales pressure, resulting in low economic benefits; in addition, the concentrated supply time cannot meet the growing market demand. Therefore, the development of new, early maturing, high-yielding pear cultivars has become one of the main goals of pear breeding programs [5].

In recent years, a number of representative early maturing cultivars have been bred, such as ‘Sucui 1’ and ‘Cuiyu’ from China, ‘Starkimson’ of America, ‘Madeleine’ of France, ‘Samlitoto’ of Italy, etc. [4,6,7,8]. Compared to late-maturing cultivars, embryos from early maturing cultivars have shorter fruit development periods and a lower seed viability and are less mature [9]. The low-percentage seed germination rate and poor regeneration limit the breeding of very early maturing pear cultivars. In vitro culture represents a powerful tool that has been widely applied in fruit breeding programs, especially for the hybridization of early maturing cultivars [10]. By culturing immature embryos or seeds on a nutrient medium under aseptic conditions, the germination rate and plant production are improved, greatly facilitating the introduction of superior early maturing cultivars [11,12,13,14]. Using this technique, successful cases have been reported regarding the introduction of early maturing cultivars of grapes, peaches, and sweet cherries [15,16,17]. However, the application of in vitro culture methods to the breeding of very early maturing pear cultivars is still limited.

Although in vitro culture is an effective way to rescue immature embryos and enhance seed germination, the success of this procedure depends on many factors, such as the composition of the growth medium, the tissue developmental stage, and the cultivation conditions [18,19,20]. Selecting an appropriate nutrition medium is one of the most important steps in tissue culture [21]. Murashige and Skoog [22] medium is the most widely used nutrient medium in embryo and seed culture. In addition, Gamborg’s B5 [23], Knopp and woody plant (WP) [24], and White [25], etc., media have also been determined as ideal for in vitro seed germination and seedling development. For example, the germination rate of early maturing grape embryos was significantly increased by culturing them in White medium [26]. Different plants have different dormancy mechanisms to adapt to the environment; generally, a period of cold treatment prior to germination is required to break seed dormancy of some plants. Hence, the temperature and duration of stratification are other key factors for the success of in vitro seed culture [20,27,28]. Studies have shown that the optimal temperature and duration of cold stratification vary greatly among tissues and species, and the stratification duration for pear seeds is 40 to 50 days [29,30]. Since the dormancy level is also affected by maturity, a longer cold treatment time may be required for early maturing cultivars with incomplete embryo developed seeds. Therefore, to obtain a better germination rate for early maturing pear cultivars, it is necessary to determine the optimal culture medium and culture conditions [31].

‘Pearl Pear’ (*Pyrus pyrifolia*) is a very early maturing variety derived from interspecies crossing between the Japanese sand pear variety ‘Yakumo’ (*Pyrus pyrifolia*) and the French early maturing pear variety ‘Beurre Gifard’(*Pyrus communis*) [32,33]. It matures in early summer and has a fruit growth period of only 73 days, representing an excellent parent material for breeding potential super-early maturing progenies [34]. However, during the breeding process, we found that the germination rate of ‘Pearl Pear’ seeds is relatively low, at less than 20%. Pearl Pear germination may be affected by factors such as incomplete embryo development, a low seed content, a lack of sufficient low-temperature conditions, and the physical limitations of the seed coat, which often limit its application in breeding programs [34]. In this study, we investigated the effects of the culture medium, the temperature and duration of cold stratification, and the seed coat on the germination percentage and seedling vigor of ‘Pearl Pear’ hybrids. Our research aims to establish an efficient in vitro seed germination protocol for early maturing pears seeds, which can further promote the breeding and selection of new early maturing pears or other fruit cultivars.

## 2. Materials and Methods

### 2.1. Plant Materials and Sterilization

Experiments were carried out on the hybrid progenies of ‘Pearl Pear’. Fruits were collected in June 2022 from the fruit germplasm nursery at the Jiangsu Academy of Agricultural Sciences in Nanjing, China. The seeds were separated from the equatorial line of the fruits using a knife. Impurities and low-vigor seeds were removed as much as possible. The seeds were then washed and collected in a clean, sealed collection tube. Further analyses were then conducted on an ultra-clean bench. The seeds were soaked in 70% ethanol for 1 min, then washed with sterile distilled water and soaked in 2% sodium hypochlorite for 8 min. After rinsing in sterile distilled water 3–5 times, they were mixed and left for 3 min to fully remove the residual sodium hypochlorite. After cleaning, the seeds were transferred to sterilized filter paper, which absorbed the excess water. Then, the seeds were placed in a triangular flask containing the culture medium using tweezers, with 5–6 seeds placed in each bottle. In addition, a portion of the seeds ware taken out after the fruit ripens, rinsed with water to remove surface sugar and residue, placed on filter paper to absorb moisture, and dried at room temperature for two weeks.

### 2.2. Effects of Different Cold Temperature Treatment Methods on Seed Germination

Mature ‘Pearl Pear’ fruits were collected and the seeds were removed and air-dried at 25 °C for later use. Seeds were stored at a cold temperature of 4 °C for 60 days using different methods: sand stratification, wet filter paper, an in vitro method, and fruit storage; the corresponding control seeds were treated at a normal temperature (25 °C). The sand stratification method and the wet filter paper treatment involve cleaning and drying seeds extracted from the fruit, subjecting them to water absorption treatment, and then placing them in moist sand or a petri dish filled with wet filter paper, respectively. The in vitro method involved the removal the seeds from the fruit in clean bench, disinfecting and cleaning them, then placing the seeds in a triangular flask containing MS culture medium for cold-temperature treatment. The fruit storage method means storing the fruit directly in a small cold storage of 4 °C (±0.2 °C). The four cold temperature treatment methods were shown in Figure 1. After the treatment, the seeds were transferred to a light culture room at 25 °C for germination, light duration is daytime:darkness = 12 h:12 h, the light intensity was set to 1000 lux, and seed germination was quantified after 12 days.

In order to explore the effects of different durations and cold temperature treatment methods on seed germination, two treatment groups were set up in the experiment: a seed preservation group on MS medium and a fruit preservation group. For the seed preservation group, seeds were sterilized and incubated on MS medium at 4 °C for 0, 25, 50, 75, 100, or 125 days in the dark, and five seeds were placed in each triangular flask containing MS medium. After cold stratification, the seeds were left to germinate at 25 °C with 12 h of light and 12 h of darkness. For the fruit preservation group, fruits were directly incubated at 4 °C for 0, 25, 50, 75, 100, or 125 days in the dark. After cold stratification, the seeds were separated from fruits and disinfected before being placed on MS culture medium, then left to germinate under the same conditions as the seed preservation group. After two weeks of cultivation, seed germination was quantified, and the rooting and cotyledon extension rates were measured for both groups.

### 2.3. Effect of Nutrient Media on Seed Germination and Seedling Establishment

Five different media (MS, C167, White, G398, and M519) were selected for this study [22,24,25,26], all of which were supplemented with 3% sucrose and 0.7% agar, the pH of the media was adjusted to 5.7. The fully sterilized seeds were transferred to a 150 mL triangular bottles containing 40 mL different media, with 5 seeds placed in each bottle to facilitate seed growth and subsequent statistical analyses. The isolated seeds were incubated at 4 °C for 100 days, then placed on plates under aseptic conditions. The bottles were maintained in a culture room at 25 °C with 12 h of light and 12 h of darkness. On days 6, 12, and 18 of cultivation, seed germination was quantified, and the rooting and cotyledon extension rates were measured.

### 2.4. Effect of Seed Coat Removal on Germination

For the fruit preservation group, the seeds were separated from fruits on an ultra-clean workbench after 100 days of cold stratification, and apical seed coat removal (ASR), middle seed coat removal (MSR), lateral seed coat removal (LSR), and full seed coat removal (FSR) were performed with a scalpel in a super clean bench, respectively (Figure 2). These four methods consist of the dissection and removal of the apical (near the radicle), middle (the upper 1/2 of the cotyledons), lateral portions, and entire seed coat, respectively. Seeds subjected to these different procedures were placed into a new triangular flask containing White medium, which was then placed in a 25 °C culture room to observe seed germination. Germination tests were subsequently conducted using the methods described below.

### 2.5. Statistical Analysis

All studies were performed using a completely randomized design with three replications per treatment and 50 seeds per replication. The germination percentage was calculated as (number of seeds showing 2 mm radicle emergence after culture/total number of seeds cultured) × 100%. The root length was measured using a vernier caliper. The rooting rate was calculated as (number of seeds with root length over 1 cm/total number of seeds cultured) × 100%. The cotyledon extension rate was calculated as (number of fully expanded seeds of cotyledons/total number of seeds cultured) × 100%. Data were analyzed using SPSS 17.0. The significance of differences was analyzed by an analysis of variance (ANOVA) and the significance level was set at *p* < 0.05.

## 3. Results

### 3.1. Effects of Different Cold Treatments on Seed Germination of ‘Pearl Pear’

To break the seed dormancy, the seeds were treated at 4 °C for 60 days using different methods such as sand stratification, wet filter paper, in vitro method, and fruit storage. The results showed that the germination rate of seeds treated with in vitro method was the highest, at 43.86%, followed by the wet filter paper method, with a germination rate of 28.35%, while the germination rate of seeds treated with sand stratification method was 19.43% (Figure 3). It is worth noting that the germination rate of the fruit storage method was only 2.8%, which may be due to the limitation of the seed coat because it has physical limitations and contains germination inhibitors such as polyphenols. Seeds treated at 25 °C did not germinate. This indicates that a low temperature is a necessary condition for pear seed germination. Different cold temperature treatment methods have different effects on promoting seed germination; a cold temperature of 4 °C for 60 days pretreatment during in vitro is an effective way to improve the seed germination of early maturing pears.

### 3.2. Effect of the Duration of Cold Stratification on Seed Germination and Growth

We pretreated both ‘Pearl Pear’ fruits and seeds at 4 °C for 25 to 125 days to determine the optimal time of cold stratification for the germination of early maturing pear seeds. Then, all seeds were germinated in MS culture medium. The results showed that the germination rate of the seed preservation group significantly improved after 25 days of cold temperature pretreatment and continued to increase with prolonged treatment times (Figure 4A). The germination rate reached the highest (84.51%) at 100 days of treatment, but slightly decreased at 125 days. For the fruit preservation group, the germination rate also improved after cold-temperature pretreatment and peaked on day 100 but was significantly lower than the seed preservation group (6.89% vs. 84.51%, *p* = 0.03). For the seed preservation group, we further investigated the effects of the cold temperature pretreatment duration on rooting and cotyledon elongation (Figure 4B). The results showed that both the rooting rate and the cotyledon elongation rate increased with treatment duration and peaked at 100 days (75.51% and 65.91%), exhibiting a similar trend to the seed germination rate. Therefore, our finding demonstrates that cold stratification (4 °C) for 100 days is best for the germination and growth of early maturing pear seeds.

### 3.3. Effects of Nutrient Medium on Seed Germination and Seedling Establishment

To explore the optimum culture medium for seed germination, five different culture media were used in the germination of early maturing pear seeds. On day 6, pear seeds in White and M519 culture media had high germination percentages of 72.98% and 72.96%. On day 12, the germination rate in White medium increased to 83.62%, and, among the rest of the media, the C167 culture medium had a germination percentage of over 80%. On day 18, the germination rate in White medium remained the highest, reaching 86.54%, followed by in vitro culture, 83.36%, and C167 culture medium, 81.73% (Figure 5A).

In addition, we also observed that the rooting and cotyledon elongation rates were higher on White culture medium than the other media. After different media treatments, the root growth rate and cotyledon elongation rate of hybrid ’Pearl Pear’ seeds increased with time. On the 12th day, the culture medium with the highest seed rooting rate was White medium, which was as high as 70.85%, significantly higher than the 54.9% on MS culture and the 50.72% on M519 culture (*p* < 0.05). The rooting rates of G398 and C167 media were relatively poor, reaching 40.7% and 31.15% on the 12th day, respectively (Figure 5B). In addition, the extension of cotyledons treated with different media was recorded after 6, 12, and 18 days of cultivation at 25 °C. The results showed that with the increase in treatment duration, the cotyledon elongation rate continued to increase, and the germination potential remained relatively high between 12 days and 18 days. On the 12th day, the cotyledon elongation rate in the White treatment group was as high as 72.98%, significantly higher than the other treatments. On the 18th day, the cotyledon elongation rate for each treatment group reached the highest level. Seeds treated with White and MS media exhibited better elongation, with elongation rates of 76.51% and 75.91%, respectively, followed by seeds treated with C167, M519, and G398, with elongation rates of 68.21%, 58.43%, and 50.94%, respectively (Figure 5C).

According to the results of the seed germination, root growth, and cotyledon extension of ‘Pearl Pear’s seeds on different mediums, we found that the White medium had significantly better effects on promoting seed germination and seedling growth than other medium treatments (*p* < 0.05). Therefore, it was concluded that the best medium conditions for seed germination and growth promotion of ‘Pearl Pear’ followed the order White > MS > C167 > G398 > M519; we recommend using White culture medium for the germination of early maturing pear seeds. The seeding growth on each medium is shown in Figure 6, and the observation time is on the 18th day after transfer to a 25 °C culture chamber.

### 3.4. Effect of Different Seed Coat Removal Methods on Germination and Growth Rates

In experiments on the effects of different low-temperature treatments on seed germination, after different durations of low-temperature treatments, it was found that even when the different cold-temperature treatments were applied for the same duration, the germination rate of hybrid offspring seeds preserved in fruit was much lower than those treated in the culture media. We speculate this may be due to the physical limitations of the seed coat. Therefore, we conducted seed coat removal experiments in the seed preservation group. Among the four different methods of seed coat removal, the FSR treatment had the highest germination rates from day 6 to day 18, followed by the MSR treatment. On day 18, the germination rates reached 87.7% and 85.2% in the FSR and MSR groups, which were significantly higher than the control group (6.7%) (Table 1).

Although the FSR treatment led to the highest germination rates among the four methods, it is also more complex and time consuming, and the contamination risk is greatly increased during the operation process. Hence, MSR treatment might be the ideal way to improve the germination rate of the fruit preservation group. Thus, we investigated the effects of the MSR treatment on rooting and cotyledon elongation. The results indicated that the rooting and cotyledon elongation rates increased to 42.3% and 46.2% on day 18, which were significantly higher than the control group (Table 2). In summary, cold temperature pretreatment with seed coat removal can be an effective way to improve the germination, rooting, and cotyledon elongation rates of fruit seeds. The seed germination rate after MSR was slightly lower than that after FSR, but the operation is simple and has little impact on seed growth after germination. Additionally, the application of a culture medium in the experiment can supply the nutrients needed for germination and growth. Therefore, considering all these factors, the middle cutting treatment is the best method to remove ‘Pearl Pear’ seed coats.

## 4. Discussion

The breeding of early maturing pear varieties has always been a challenging topic in pear breeding. Currently, the proportion of early maturing pear varieties in China is relatively small, and there is an urgent need to cultivate more early maturing pear varieties to fill the gap in pear maturity. The extremely early maturing variety ‘Pearl Pear’ is an excellent parent material for breeding early maturing pears, but the problem of low germination in the seeds of hybrid offspring of ‘Pearl Pear’ is common, which limits the seedling formation of hybrid offspring of ‘Pearl Pear’ [34]. Therefore, studying the technology of seed germination and seedling formation in hybrid offspring of ‘Pearl Pear’ is of great significance for the breeding of early maturing pear varieties. Seed dormancy is a key factor impeding the seed germination of early maturing cultivars, and is often affected by physiological processes or/and structural characteristics, and factors such as temperature, moisture, and nutrients in the external environment can affect the dormancy state of seeds and regulate their germination [35]. For seeds with physiological dormancy (PD), cold stratification treatment is an effective way to stimulate seed germination [36]. In our experiment, four different cold-temperature treatments were used to treat the seeds of ‘Pearl Pear’ at cold temperature, while the seeds without cold temperature treatment did not germinate, which indicated that cold temperature was necessary to break the germination of pear seeds. Different treatments have different effects on promoting the seed germination of ‘Pearl Pear’, among which the seed germination rate of the in vitro method (MS medium) is the highest, which is 43.86%, while the germination rate of the conventional sand stratification treatment method is 19.43%. This may be because the seeds of ‘Pearl Pear’ contain less substances, and adding exogenous nutrients can provide nutritional sources for the seed and promote germination.

In addition, we treated both seeds and fruits separately under low-temperature conditions (4 °C) and found that the seed germination of early maturing hybrids was significantly improved in the seed preservation group (Figure 3). For the fruit preservation group, we found that seed germination rates were much lower than those of the seed preservation group. After dissecting the fruit, we observed that the seed coat color had changed from white to dark brown, indicating a complete post-maturing process [37]. We speculated that the fruit pulp may weaken the seed’s ability to perceive changes in the external environment, which, together with the isolation of oxygen, limits seed germination [38,39]. For the seed preservation group, the pear seeds reached their highest germination rates of 85% after 100 days of cold stratification. The cold stratification treatment is consistent with the natural conditions that seeds endure before germination, and the optimal period of cold stratification varies between and within species [40,41]. In general, 30 to 60 days of cold treatment can overcome the embryo dormancy problem in many plants [42,43,44]. For instance, cold stratification for a period of 60 days can completely break the seed dormancy of Callery pears (*Pyrus calleryana* Decne.) [45], and it only takes about 30 days for the late-maturing ‘Huangguan’ pear variety (*Pyrus bretschneideri* Rehd.) to reach 90.5% seed germination [46]. The dormancy release and germination of seeds are closely related to external environmental conditions and are the result of long-term natural evolution. Low temperature is an important factor affecting the dormancy process of pear seeds, affecting the changes of related enzyme activities in embryos, further affecting the metabolic activities of seeds such as water absorption, respiration, and nutrient transformation, and finally affecting the germination and growth of seedling [36]. For early maturing pear cultivars, our study showed that longer periods of cold stratification—at least 3 months—are necessary for improved seed germination.

Improving seed germination and seedling establishment is a key step to accelerate the breeding of new early maturing pear cultivars [33]. In vitro culture methods are widely used to improve the poor seed germination of early maturing cultivars in many fruit species [47,48,49]. In this study, high germination rates were also achieved by using the in vitro seed culture method for the very early maturing pear cultivar hybrids. Generally, MS medium is the most widely used medium for embryo germination and seedling growth [50]. However, nutritional requirements vary among different species and seed types. For instance, it was found that Hyponex medium had the best effect on the germination of hybrid seeds from immature capsules [51], and Meng et al. greatly improved the rooting rates of ‘Dangshan’ pear transgenic seedlings by culturing in ASH and PG culture media [52]. Among the five media applied to ‘Pearl Pear’ hybrids, we found that MS and White medias are more effective in promoting seed germination, while for rooting and cotyledon extension after seed germination, White medium had the best effect. MS medium is characterized by a high salt concentration; some studies have shown that a high salinity has negative effects on germination percentages, seedling growth, chlorophyll content, etc. [53]. Different media types contain different nutrients and have different effects on seed germination and growth. The analysis of the nutrients in each culture demonstrated that the White medium contained fewer inorganic salts and had an increased boron content, which is more conducive to seed germination and seedling rooting of early maturing pear cultivars.

In addition to embryo dormancy, seed-coat-imposed dormancy is another important mechanism that restricts seed germination via impermeability, mechanical resistance, or the supply of inhibitory substances [54]. Studies have found that the seed coat plays an important role in maintaining pear seed dormancy, and removing the seed coat can significantly improve pear seed germination [55,56]. Some studies have suggested that the pear seed coat contains inhibitory substances like ABA hormones that inhibit seed germination [57,58]. In this study, we found that all the different seed coat removal methods increased the seed germination of ‘Pearl Pear’ hybrids to varying degrees, indicating that the seed coat may also have mechanical resistance to pear seed germination. Although the germination rates reached 87.7% with full seed coat removal, this method is complicated, time-consuming, and greatly increases the contamination risk. In contrast, middle seed coat removal led to a high germination of 85.2%, but very low contamination. It promoted cotyledon emergence more than lateral seed coat removal and did not impair radicle-like apical seed coat removal. Hence, we suggest that middle seed coat removal is an ideal method to relieve the mechanical restriction of the seed coat and increase the seed germination rate of pears.

## 5. Conclusions

Seeds of early maturing pear cultivars are affected by factors such as incomplete embryo development and seed coat limitations, exhibiting strong dormancy, resulting in a low germination rate of the seeds, which limits the application of ‘Pearl Pear’ in breeding. The factors that affect the germination of early maturing pear seeds have been revealed under culture conditions, and a highly efficient protocol for the germination of early maturing pear seeds has been established. The results demonstrate that certain treatments, such as sufficient cold stratification, a suitable culture medium, and complete seed coat removal, have highly positive effects on stimulating germination. We determined that a 100-day period of cold stratification on White medium cultivation are optimal for the germination and growth of early maturing pear seeds. In addition, using the middle cutting treatment can greatly improve the seed germination rate. These findings are of importance for the increased utilization of very early maturing pear cultivars as parental materials for crossing and for early maturing variety breeding programs. The disadvantage of this study is that the time from seed germination was still very long, even after the use of seed skin removal and low-temperature treatment in the medium. The next step will be combining measures such as different temperatures and hormone treatments to break the dormancy of ‘Pearl Pear’ seeds in a shorter time and improve the germination rate of the seeds.

## Figures and Tables

**Figure 1 plants-12-04120-f001:**
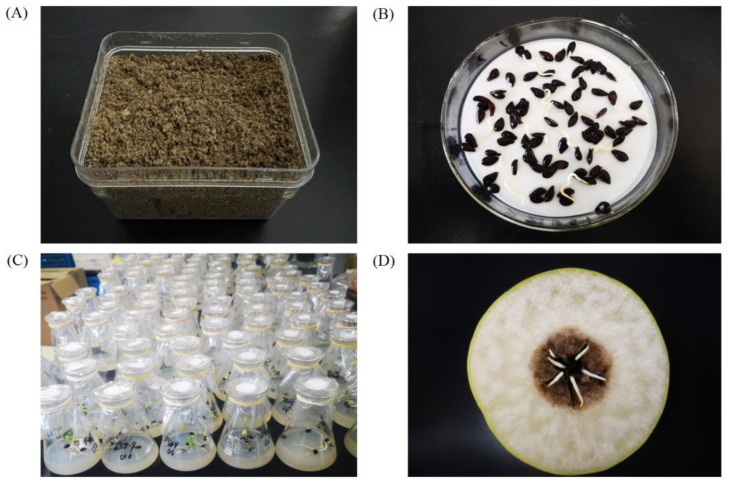
Four different cold temperature treatment methods of early maturing *Pyrus perifolia* seeds. (**A**) Sand stratification; (**B**) wet filter paper; (**C**) in vitro method; (**D**) fruit storage. All the seeds in the four methods were placed at a cold temperature of 4 °C for 60 days 2.3. Effects of cold temperature treatment duration on seed germination.

**Figure 2 plants-12-04120-f002:**
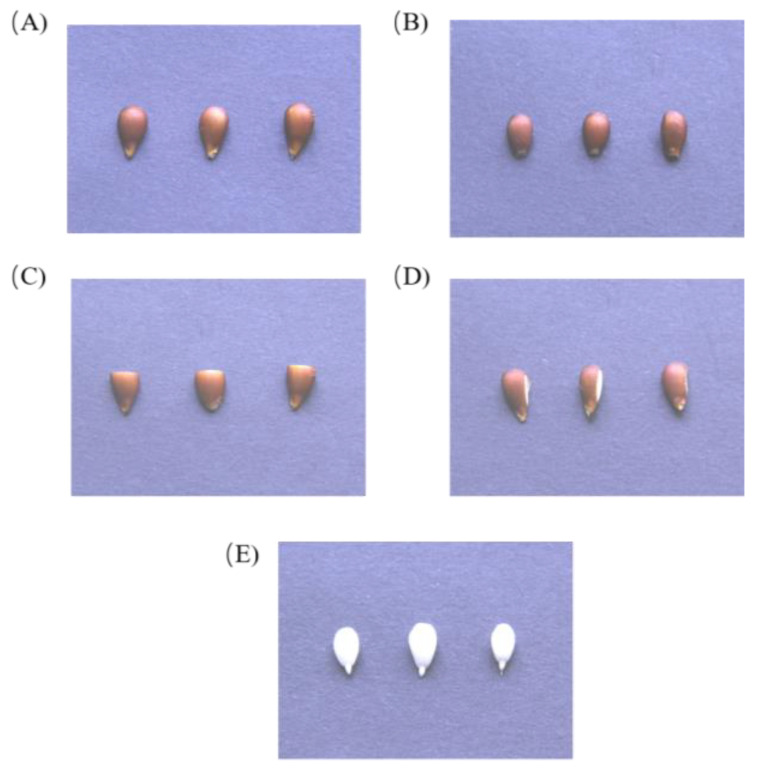
Four different seed coat removal methods. (**A**) Control group; (**B**) apical seed coat removal (ASR); (**C**) middle seed coat removal (MSR); (**D**) lateral seed coat removal (LSR); (**E**) full seed coat removal (FSR).

**Figure 3 plants-12-04120-f003:**
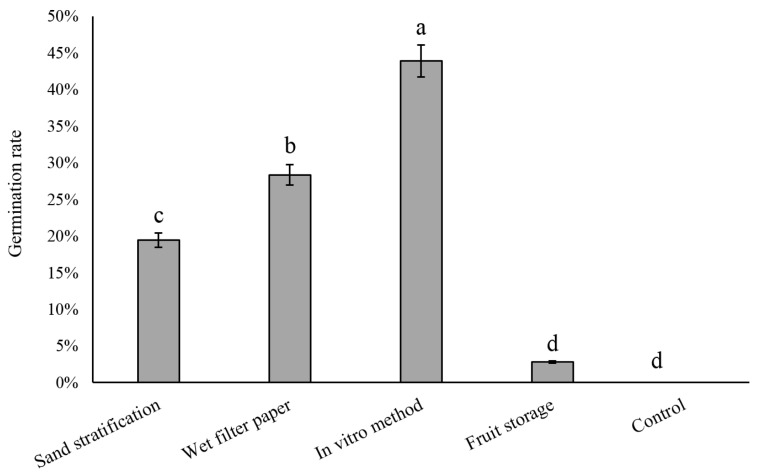
Germination rates of early maturing pear seeds with different cold treatments. The error bars indicate the standard deviations of three replications. The different letters above the bars indicate significant differences between samples according to one-way ANOVA (Duncan’ test, *p* < 0.05).

**Figure 4 plants-12-04120-f004:**
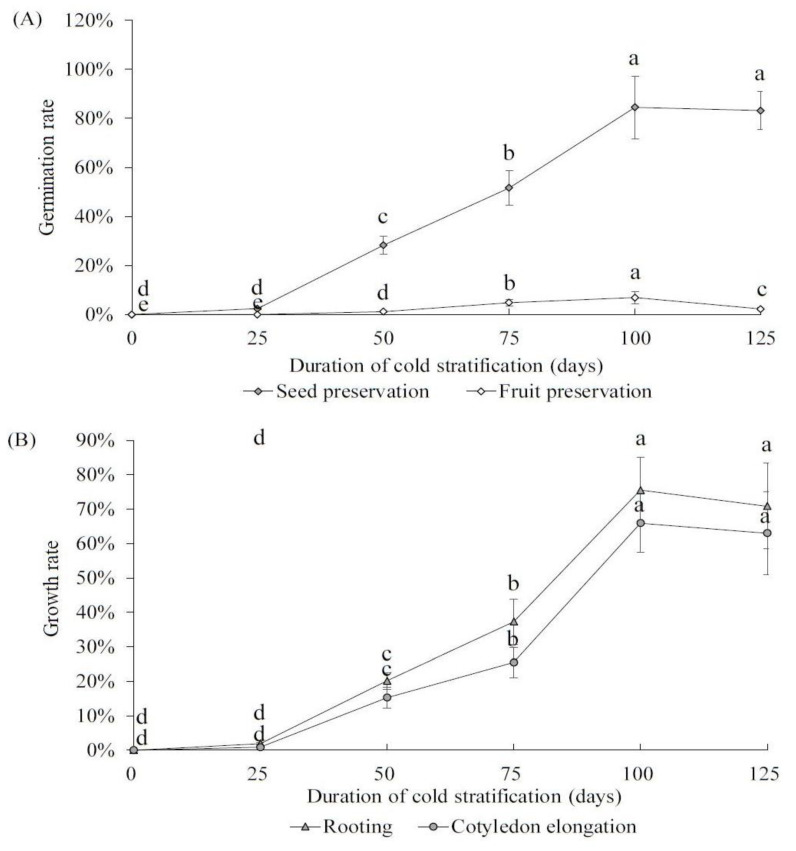
(**A**) Germination rates of early maturing pear fruits and seeds with 4 °C pre-treatment for 25 to 125 days; (**B**) rooting rates and cotyledon elongation rates of early maturing pear fruits and seeds with 4 °C pre-treatment for 25 to 125 days. The error bars indicate the standard deviations of three replications. The different letters above the bars indicate significant differences between samples according to one-way ANOVA (Duncan’ test, *p* < 0.05).

**Figure 5 plants-12-04120-f005:**
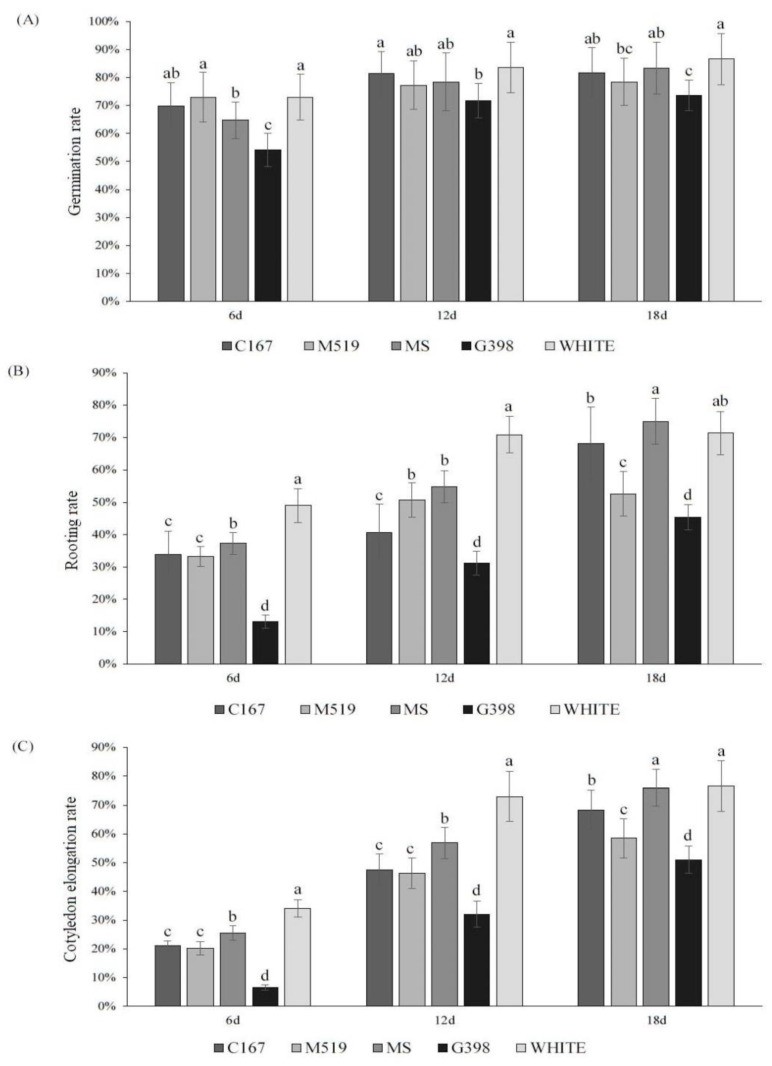
(**A**) Germination rates of early maturing pear seeds on seven different culture media; (**B**) rooting rates of early maturing pear seeds on seven different culture media; (**C**) cotyledon elongation rates of early maturing pear seeds on five different culture media. The error bars indicate the standard deviations of three replications. The different letters above the bars indicate significant differences between samples according to one-way ANOVA (Duncan’ test, *p* < 0.05).

**Figure 6 plants-12-04120-f006:**
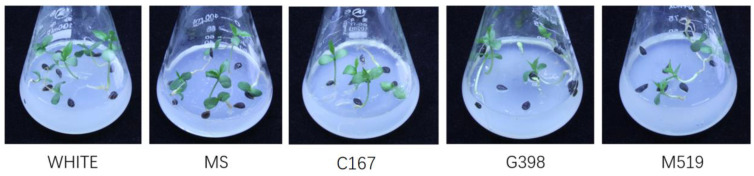
Seedling growth on different medium conditions after 18 days of cultivation. Five different mediums media (MS, C167, White, G398, and M519) were selected, all the seeds were incubated at 4 °C for 100 days.

**Table 1 plants-12-04120-t001:** Germination rates of four seed coat removal methods. Seeds were processed with apical seed coat removal (ASR), middle seed coat removal (MSR), lateral seed coat removal (LSR), and full 135 seed coat removal (FSR), respectively. The different letters above the bars indicate significant differences between samples according to one-way ANOVA (Duncan’ test, *p* < 0.05).

	Germination Rate (%)				
Days after Incubation	Control	ASR	MSR	LSR	FSR
6	4.88 ± 0.38 d	23.52 ± 3.15 c	28.83 ± 3.17 c	45.75 ± 6.58 b	53.74 ± 4.57 a
12	10.34 ± 1.12 d	35.14 ± 4.16 c	47.16 ± 5.86 b	69.34 ± 7.12 a	75.22 ± 8.26 a
18	7.32 ± 0.68 c	41.67 ± 5.19 b	51.75 ± 6.42 b	85.22 ± 6.18 a	87.73 ± 7.62 a

**Table 2 plants-12-04120-t002:** Rooting and cotyledon elongation rates of the middle seed coat removal (MSR) method. The different letters above the bars indicate significant differences between samples according to one-way ANOVA (Duncan’ test, *p* < 0.05).

	Rooting Rate (%)		Cotyledon Elongation Rate (%)	
Days after Incubation	Control	MSR	Control	MSR
6	0.00 ± 0.00 b	19.61 ± 2.16 a	0.00 ± 0.00 b	16.37 ± 2.43 a
12	3.92 ± 4.18 b	37.79 ± 3.76 a	3.45 ± 2.78 b	32.79 ± 3.71 a
18	4.55 ± 3.96 b	42.31 ± 5.12 a	4.17 ± 5.49 b	46.15 ± 5.68 a

## Data Availability

All data have been included in the main text.

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
