# Peer review of "Seed Germination and Growth Improvement for Early Maturing Pear Breeding"

_plants, 2023, doi:10.3390/plants12244120_

Round 1

Reviewer 1 Report

Comments and Suggestions for Authors

Dear Authors,
This manuscript describes quite interesting experiments on in vitro germination improvement. It has more practical applications than scientific novelty.
The weakest part is the manner that which those experiments are described. It is quite difficult to understand for the reader. I suggest very precisely correct Materiał and methods part, with the help of an experienced service or person.
Because the research was composed of several stages it is not clear when was done each of the stages. I think that a clear Graphic describing the experiment plan would make this research more understandable.
After correcting Mat&met you can correct the Results and the other parts of the manuscript.

Comments on the Quality of English Language

Please be conscious about the grammatic as you often mix tenses from present to past. The English language needs extensive editing because is very difficult to understand.

Author Response

Thank you very much for taking the time to review this manuscript and put forward your valuable opinions, which are very valuable for improving the quality of this manuscript . Please find the detailed responses below and the corresponding in track changes in the re-submitted files. The list of responses to the comments will be uploaded as a file; namely, the covering letter, with our revised manuscript ‘Manuscript -plants-2678738-revision’. If you have any suggestions regarding aspects of the manuscript that need further improvement, please contact me.

Reviewer 2 Report

Comments and Suggestions for Authors

The study conducted by Jialiang Kan and colleagues explores the vital issue of improving seed germination and growth in early-maturing pear breeding. Early-maturing pear cultivars are essential for horticultural practices, as they serve as valuable parental material for breeding, but their low seed germination and immature embryos have been limitations. The authors focused on the 'Pearl Pear' cultivar and investigated the influence of various factors, including cold stratification, culture medium, and seed coat removal, on seed germination and growth. This review report aims to assess the quality of the study and its implications for early-maturing pear breeding.

Introduction:

The introduction concisely outlines the significance of the research problem. It effectively highlights the challenges associated with early-maturing pear breeding, such as low seed germination and immature embryos. The choice of 'Pearl Pear' as the subject of investigation is justified, as it is a very early-maturing cultivar with the potential to address these challenges. The objectives of the study are well-defined, emphasizing the need to enhance seed germination in early-maturing pear cultivars to facilitate breeding.

Methods:

The research methods are described with clarity. The study employs cold stratification at 4°C, various culture media, and seed coat removal techniques. The use of White medium and the specific method of middle seed coat removal (MSR) are detailed, providing readers with sufficient information to replicate the experiments. The methodology is appropriate for the research objectives, and the rationale for each technique is explained well.

Results:

The study's results are clear and well-structured. The findings show that cold stratification significantly improves early-maturing pear seed germination, with a 100-day pre-treatment at 4°C leading to an impressive 84.54% germination rate on White medium. Furthermore, seed coat removal, particularly middle seed coat removal (MSR), is demonstrated to enhance germination activity. These results are presented in a format that is easy to interpret, with tables or figures providing visual support for the data.

Discussion:

The discussion section provides a comprehensive analysis of the results and their implications. The authors discuss the significance of their findings in the context of early-maturing pear breeding, emphasizing how these improved germination protocols will facilitate the development of new early-maturing cultivars. The choice of cold stratification and the White medium is well-justified, and the preference for middle seed coat removal (MSR) is explained logically. The study's limitations and potential areas for further research are also acknowledged.

Conclusion:

The study offers valuable insights into improving seed germination and growth in early-maturing pear breeding. The combination of cold stratification, the White medium, and middle seed coat removal (MSR) stands out as an effective protocol for enhancing germination rates. The conclusions align with the study's objectives and have practical implications for horticulturists and researchers in the field of pear breeding.

Overall Assessment:

This research report is well-structured, with a clear introduction, methods, results, discussion, and conclusion. The methods employed are appropriate, and the results are presented logically. The discussion effectively links the findings to the broader context of early-maturing pear breeding. The study's conclusions are well-supported by the results and contribute to the advancement of horticultural research in this area. This work can be considered a valuable resource for scientists and horticulturists interested in early-maturing pear cultivars.

Author Response

Dear editor, thank you for your approval of this manuscript. It seems that you have not raised too many questions about this manuscript. According to the requirements of the editorial department, we have made a lot of revisions to the content of this manuscript, and the revised manuscript is named' Manuscript-Plants-2678738-Revision'. The modified part is marked in the modification mode in word, and the main modified contents are described in detail in the file' Responsible Reviewer Comments-MDPI'. I hope the revised manuscript can get your approval again. Thank you very much for taking the precious time to review this manuscript, if you have any suggestions regarding aspects of the manuscript that need further improvement, please contact me.

3. Point-by-point response to Comments and Suggestions for Authors

Comments 1: When was Seed Harvest Day? And how long was it stored and used in the experiment? In seeds with physiological dormancy, storage method and storage period have an effect on lowering the depth of dormancy. Therefore, we need to provide information on these aspects.

Response 1: Thank you for pointing this out. We agree with this comment. Therefore, we have added relevant contents to the materials and methods.

[ In 2.1 of Materials and Methods, the collection time of seeds was increased in June of 2022. One part of the seeds was directly used in the experiment after collection, and the other part was dried at room temperature for 2 weeks after cleaning.]

Comments 2: In line 22’A 100 days of cold temperature pre-treatment’, is it the same as cold stratification?

Response 2: Thank you for pointing this out.

[There is something wrong with the expression of this sentence, and " A 100 days of cold temperature pre-treatment " has been changed to " One hundred days of cold temperature treatment ". This change can be found in line 23.]

Comments 3: [line 68Depends on the plant, not all species need dormancy. This generalization  gone too far]

Response 3: Agree. hank you for pointing this out. We agree with this comment.

This part of the discussion about breaking seed dormancy at low temperature is not rigorous, and it is changed to“Generally, a period of cold treatment prior to germination is required to break seed dormancy” This change can be found in line 74—75

Comments 4: [line 74: what does it means? you mean that embryo is in globular stage? what is seed is poorly developed?]

Response 4: Agree. Thank you for your reminder. We agree with this comment.

The description of "poorly developed" in the original manuscript is not very accurate, and " early maturing cultivars with poorly developed seeds. " has been changed to " early-maturing cultivars with incomplete embryo developed seeds ". This change can be found in line84.

Comments 5: [Material and methods: It is difficult to understand the sequence of research conducted and the environmental conditions in which it took place. I suggest authors think about the order of experiments and describe them in an understandable and grammatically correct way. Perhaps it would be easier for the reader to use a research scheme or experimental plan in chronological order and with a description of what was observed and when.]

Response 5: A Thank you for pointing this out. We agree with this comment. Therefore, we have added relevant contents to the materials and methods.

The materials and methods are supplemented to ensure that readers can easily understand the operation process of the experiment, such as adding the explanations of four different low-temperature treatment methods in Method 2.2 and adding pictures to show them. At the same time, a lot of changes have been made to the description of word tenses and sentences in the material method, hoping that this can meet your requirements.

Comments 6: [Material and methods2.2: IT IS ALL VERY CONFUSING AND NOT CLEAR AT ALL]

Response 6: Agree. Thank you for your reminder. You have put forward a lot of valuable opinions about Part 2.2 of Materials and Methods, and I have made a lot of amendments to this part.

First of all, in this part, the explanations of four different low-temperature treatment methods in the operation process are added respectively, and the effect diagrams of four different methods are added, which is more conducive to readers to understand how to operate; The temperature and time of treatment are also clearly stated; What kind of environment can promote the germination of seedlings after low temperature treatment is also explained. Finally, the grammar and expression of this part are adjusted to ensure the fluency and correctness of the language.

Comments 7: [line 124-126: what was the capacity of the bottle (ml)and capacity of the media (ml) in eachwhat was the capacity of the bottle (ml)and capacity of the media (ml) in each]

Response 7: Agree. Thank you for your reminder.

This part of the description is not sufficient, and the corresponding explanation has been added to the manuscript.The fully sterilized seeds were transferred to a 150m triangular bottles containing 40 ml different media, with 5 seeds placed in each bottle”.The changes can be found in line 157-159

Comments 8: [In graphs, the numbers on the y-axis do not need to be written to two decimal places. Everything needs fixing]

Response 8: Agree. Thank you for your reminder.

In figures 3, 4 and 5, two decimal places have been deleted from the Y-axis.

Comments 9: [line 164 : which may be due to the limitation of the seed coat --> What does “which may be due to the limitation of the seed coat” mean? It would be nice if you could explain it in more detail.]

Response 9: Agree. Thank you for your reminder.

The speculative reasons for limiting seed germination have been added in lines 202-203: ‘because it has physical limitations and contains germination inhibitors such as poly-phenols’

Comments 10: [line 224 : WHITE>MS>C167>G398>M519 --> It must be described based on statistical significance.]

Response 10: Agree. Thank you for your reminder.

The investigation and statistical results of seed germination, root growth and cotyledon elongation of hybrid offspring of Pearl pear on different media were significantly analyzed, and on this basis, the conclusions obtained were described complementarily. The changes can be found in figure 5 and line 275-277.

Comments 11: [line 260 : for seeds with MD ~~ --> This is a more common phenomenon in seeds with physiological dormancy (PD) than in MD dormancy, so this part requires correction.]

Response 11: Agree. Thank you for your reminder.

In question 18, the discussion that seeds with morphological dormancyMD are more likely to be broken by low-temperature stratification has been checked from the cited articles and corrected to "physiological dormancy(PD)".The change can be found in line 336.

Comments 12: Table 1: you have to explain those abreviations. Table has to be readible and easy to understand without text.]

Response 12: Agree. Thank you for your reminder. Therefore, we have added relevant contents in the table.

In Tables 1 and 2, the title at the top left is ''cold stratification (days)" had been changed by “Days after incubation’’

Significant analysis has been added to tables 1 and 2, and abbreviations of different cutting methods have been added to the title.

4. Response to Comments on the Quality of English Language

Point 1: line 21 : ‘could’ should be deleted, ‘improve’ should be past tense

Response 1:    ‘could’ had been deleted, ‘improve’ changed to  ‘improved’, The changes can be found in line 21 and 22.

Point 2: line 23 : ‘(could)’ should be deleted

Response 2:    ‘(could)’ had been deleted in line 24

Point 3: line 25 : ‘(MSR)’should be deleted

Response 3:    ‘(MSR)’ had been deleted In line 26

Point 4: line 31 : spp. --> not italic

Response 4:    change to italics. In line 33

Point 5: line 35 : According to ~~ --> Leave a space

Response 5:    A space has been added in front of According to. In line 37

Point 6: line 46 : of --> at

Response 6:   of change to at in line 50

Point 7: line 53 : in or on?

Response 7:    in change to on in line 57

Point 8: line 54 and 57 : maturing ~~ --> Leave a space

Response 8:    A space has been added in front of maturing In line 60 and 63

Point 9: line 58 : “is still limited. ”, what about other pearl cultivars? is this limited too?

Response 9:    change to italics

Point 10: line 65 : and White ~

Response 10:    White change to and White in line 71

Point 11: line 85 : coat. --> coat

Response 11:   coat. change to coat, in line 95

Point 12: line 94 : I think that intitution name should be in capitals

Response 12:    “Jiangsu academy of agricultural sciences”change to “Jiangsu Academy of Agricultural Sciences” ,changed in line 106

Point 13: line 98 : should or were

Response 13:    ‘should’ change to “were”

Point 14: line 99100102 : spp. --> some words should be deleted

Response 14:    (C2H5OH)、(NaOCl)、 ‘the seeds’ had been deleted

Point 15: line 140 : ‘are’ , grammar mistake. please use one tense to describe method

it apply to all text

Response 15:   ‘are’ change to “were” in line 174, At the same time, the temporal problem of the full text is checked and modified.

Point 16: line 158 : The seeds --> the seeds

Response 16:   change ‘The seeds ‘to ‘the seeds’ in line 195

Point 17: line 343 : Spp. --> spp. not italic

Response 17:   Adjust Spp. in line 450 to non-italic. In line 449

5. Additional clarifications

[The title of the article " Improvement of the seed germination and growth in early-maturing pear breeding" was changed to "Seed germination and growth improvements for early-maturing pear breeding"

The discussion and conclusion in the manuscript are expanded and enriched.

Other details in the manuscript have also been revised in the manuscript, and marks have been displayed in the review mode.]

Reviewer 3 Report

Comments and Suggestions for Authors

This manuscript contains information on improving seed germination during early maturing. This is an important research result for the breeding of new varieties.

1. line 4~5 : *; --> ??

2. line 31 : spp. --> not italic

3. line 35 : According to ~~ --> Leave a space

4. line 65 : and White ~

5. line 85 : coat. --> coat

6. When was Seed Harvest Day? And how long was it stored and used in the experiment? In seeds with physiological dormancy, storage method and storage period have an effect on lowering the depth of dormancy. Therefore, we need to provide information on these aspects.

7. line 107 : cold temperature --> At what temperature was it treated?

8. line 148 : Petri dish --> Must start with a capital letter. Corrections are needed throughout the text.

9. line 148 : 50 seeds per treatment per replication --> 50 seeds per replication

10. line 158 : The seeds --> the seeds

11. line 164 : which may be due to the limitation of the seed coat --> What does “which may be due to the limitation of the seed coat” mean? It would be nice if you could explain it in more detail.

12. In graphs, the numbers on the y-axis do not need to be written to two decimal places. Everything needs fixing.

13. Mark statistical significance letters on graphs. Also, add a description to the title of the graph. And add a description of what the error bars mean. This applies to all graphs.

14. line 179 : gradually decreased afterwards --> After day 100, there is only data for day 125. It is not appropriate to say that it is gradually decreasing.

15. line 224 : WHITE>MS>C167>G398>M519 --> It must be described based on statistical significance.

16. In Tables 1 and 2, the title at the top left is ''cold stratification (days)". This appears to be the number of days after incubation. This needs to be corrected.

17. In Tables 1 and 2, statistical significance should be indicated and interpreted based on it.

18. line 260 : for seeds with MD ~~ --> This is a more common phenomenon in seeds with physiological dormancy (PD) than in MD dormancy, so this part requires correction.

19. line 343 : Spp. --> spp. not italic

20. The cited literature part needs to be modified to fit the format of the plants as a whole. For example, many parts need to be modified, such as notation of volume, issue, etc., and upper and lower case letters.

Comments on the Quality of English Language

This manuscript contains information on improving seed germination during early maturing. This is an important research result for the breeding of new varieties.

1. line 4~5 : *; --> ??

2. line 31 : spp. --> not italic

3. line 35 : According to ~~ --> Leave a space

4. line 65 : and White ~

5. line 85 : coat. --> coat

6. When was Seed Harvest Day? And how long was it stored and used in the experiment? In seeds with physiological dormancy, storage method and storage period have an effect on lowering the depth of dormancy. Therefore, we need to provide information on these aspects.

7. line 107 : cold temperature --> At what temperature was it treated?

8. line 148 : Petri dish --> Must start with a capital letter. Corrections are needed throughout the text.

9. line 148 : 50 seeds per treatment per replication --> 50 seeds per replication

10. line 158 : The seeds --> the seeds

11. line 164 : which may be due to the limitation of the seed coat --> What does “which may be due to the limitation of the seed coat” mean? It would be nice if you could explain it in more detail.

12. In graphs, the numbers on the y-axis do not need to be written to two decimal places. Everything needs fixing.

13. Mark statistical significance letters on graphs. Also, add a description to the title of the graph. And add a description of what the error bars mean. This applies to all graphs.

14. line 179 : gradually decreased afterwards --> After day 100, there is only data for day 125. It is not appropriate to say that it is gradually decreasing.

15. line 224 : WHITE>MS>C167>G398>M519 --> It must be described based on statistical significance.

16. In Tables 1 and 2, the title at the top left is ''cold stratification (days)". This appears to be the number of days after incubation. This needs to be corrected.

17. In Tables 1 and 2, statistical significance should be indicated and interpreted based on it.

18. line 260 : for seeds with MD ~~ --> This is a more common phenomenon in seeds with physiological dormancy (PD) than in MD dormancy, so this part requires correction.

19. line 343 : Spp. --> spp. not italic

20. The cited literature part needs to be modified to fit the format of the plants as a whole. For example, many parts need to be modified, such as notation of volume, issue, etc., and upper and lower case letters.

Author Response

Thank you very much for taking the time to review this manuscript and put forward your valuable opinions, which are very valuable for improving the quality of this manuscript . Please find the detailed responses below and the corresponding in track changes in the re-submitted files. The list of responses to the comments will be uploaded as a file; namely, the covering letter, with our revised manuscript ‘Manuscript -plants-2678738-revision’. If you have any suggestions regarding aspects of the manuscript that need further improvement, please contact me.

3. Point-by-point response to Comments and Suggestions for Authors

Comments 1: When was Seed Harvest Day? And how long was it stored and used in the experiment? In seeds with physiological dormancy, storage method and storage period have an effect on lowering the depth of dormancy. Therefore, we need to provide information on these aspects.

Response 1: Thank you for pointing this out. We agree with this comment. Therefore, we have added relevant contents to the materials and methods.

[ In 2.1 of Materials and Methods, the collection time of seeds was increased in June of 2022. One part of the seeds was directly used in the experiment after collection, and the other part was dried at room temperature for 2 weeks after cleaning.]

Comments 2: [line 107 : cold temperature --> At what temperature was it treated?]

Response 2: Agree. We have added the content about temperature to emphasize this point. Chanced the “teeated at cold temperature” to “stored at a cold temperature of 4 ℃’ in line 124

Comments 3: [line 148 : Petri dish --> Must start with a capital letter. Corrections are needed throughout the text.]

Response 3: Agree.

The description is wrong, and Petri dish is deleted on line 184 after confirmation.

Comments 4: [line 164 : which may be due to the limitation of the seed coat --> What does “which may be due to the limitation of the seed coat” mean? It would be nice if you could explain it in more detail.]

Response 4: Agree. Thank you for your reminder.

The speculative reasons for limiting seed germination have been added in lines 202-203: ‘because it has physical limitations and contains germination inhibitors such as poly-phenols’

Comments 5: [In graphs, the numbers on the y-axis do not need to be written to two decimal places. Everything needs fixing]

Response 5: Agree. Thank you for your reminder.

In figures 3, 4 and 5, two decimal places have been deleted from the Y-axis.

Comments 6: [Mark statistical significance letters on graphs. Also, add a description to the title of the graph. And add a description of what the error bars mean. This applies to all graphs.]

Response 6: Agree. Thank you for your reminder.

A significance analysis was conducted on the investigation and statistical results of seed germination, root growth, and cotyledon elongation of hybrid offspring of Pear Pear on different culture media, and supplementary descriptions were provided based on this.

Comments 7: [line 179 : gradually decreased afterwards --> After day 100, there is only data for day 125. It is not appropriate to say that it is gradually decreasing.]

Response 7: Agree. Thank you for your reminder.

The description has been changed to' The germination rate reached to the highest (84.51%) at 100 days of treatment, but slightly decreases at the 125 days' in line 220, which is more accurate.

Comments 8: [line 224 : WHITE>MS>C167>G398>M519 --> It must be described based on statistical significance.]

Response 8: Agree. Thank you for your reminder.

The investigation and statistical results of seed germination, root growth and cotyledon elongation of hybrid offspring of Pearl pear on different media were significantly analyzed, and on this basis, the conclusions obtained were described complementarily. The changes can be found in figure 5 and line 275-277.

Comments 9: [In Tables 1 and 2, the title at the top left is ''cold stratification (days)". This appears to be the number of Days after incubation. This needs to be corrected.]

Response 9: Agree. Thank you for your reminder.

In Tables 1 and 2, the title at the top left is ''cold stratification (days)" had been changed by “Days after incubation’’

Comments 10: [In Tables 1 and 2, statistical significance should be indicated and interpreted based on it.]

Response 10: Agree. Thank you for your reminder.

Significant analysis has been added to tables 1 and 2, and abbreviations of different cutting methods have been added to the title.

Comments 11: [line 260 : for seeds with MD ~~ --> This is a more common phenomenon in seeds with physiological dormancy (PD) than in MD dormancy, so this part requires correction.]

Response 11: Agree. Thank you for your reminder.

In question 18, the discussion that seeds with morphological dormancyMD are more likely to be broken by low-temperature stratification has been checked from the cited articles and corrected to "physiological dormancy(PD)".The change can be found in line 336.

Comments 12: [The cited literature part needs to be modified to fit the format of the plants as a whole. For example, many parts need to be modified, such as notation of volume, issue, etc., and upper and lower case letters.]

Response 12: Agree. Thank you for your reminder.

The format of the reference is modified according to the requirements of Plants.

4. Response to Comments on the Quality of English Language

Point 1: line 4~5 : *; --> ??

Response 1:    The * in the author has been deleted. In line 5

Point 2: line 31 : spp. --> not italic

Response 2:    change to italics. In line 33

Point 3: line 35 : According to ~~ --> Leave a space

Response 3:    A space has been added in front of According to. In line 37

Point 4: line 65 : and White ~

Response 4:    White change to and White in line 71

Point 5: line 85 : coat. --> coat

Response 5:   coat. change to coat, in line 95

Point 6: line 148 : 50 seeds per treatment per replication --> 50 seeds per replication

Response 6:   change ‘50 seeds per treatment per replication ‘to 50 seeds per replication in line 184

Point 7: line 158 : The seeds --> the seeds

Response 7:   change ‘The seeds ‘to ‘the seeds’ in line 195

Point 8: line 343 : Spp. --> spp. not italic

Response 8:   Adjust Spp. in line 450 to non-italic. In line 449

5. Additional clarifications

[The title of the article " Improvement of the seed germination and growth in early-maturing pear breeding" was changed to "Seed germination and growth improvements for early-maturing pear breeding"

The discussion and conclusion in the manuscript are expanded and enriched.

Other details in the manuscript have also been revised in the manuscript, and marks have been displayed in the review mode.]

Round 2

Reviewer 1 Report

Comments and Suggestions for Authors

Dear Authors,
The manuscript has been thoroughly revised. This significantly improved the quality of the text and the form of presentation. I suggest replacing the name of the MS MEDIUM method in the text with "in vitro cultivation" - what is the correct name for the growing method of in vitro cultures. Other notes in the text.

Comments on the Quality of English Language

I have a general comment regarding grammar, especially the use of tenses - and I suggest that you revise the article carefully in this respect.

Author Response

Thank you very much for taking the time to review this manuscript and put forward your valuable opinions, which are very valuable for improving the quality of this manuscript . We had made further revisions to this manuscript based on your valuable feedback, please find the detailed responses below and the corresponding in track changes in the re-submitted files. Our revised manuscript  named ‘Manuscript -plants-2678738-1202’. If you have any suggestions regarding aspects of the manuscript that need further improvement, please contact me.

3. Point-by-point response to Comments and Suggestions for Authors

Comments 1: line 23, ‘treatment and germination’ ,not clear.

Response 1: Thank you for pointing this out. We agree with this comment.

[Annotations have been added as required, and they have been revised to” treatment in 4 and in vitro germination” .]

Comments 2: [line 30 : Seed germination, and early-maturing occurs in the title. Acoording to standards words used in title can not be key words]

Response 2: Agree. Thank you for pointing this out.

Chanced the Keywords to: “Pyrus pyrifolia; seed dormancy; cold temperature treatment ;germination;  dormancyseedling growth’ in line 30-31

Comments 3: [line125 : do you mean "in vitro method"? because MS medium is no a name of the method. it is just a name for the medium used, which is just one part of the method]

Response 3: Agree. Thank you for your reminder.

The description of 'MS medium' is inaccurate and has been changed to 'in vitro method'. Corresponding modifications have also been made to Figures 1, 3, and the content in the manuscript.

Comments 4: [line 128 : it is best to use the past tense (Simple Past) to describe the methodology. this note applies to the entire text - especially Mat& Met, because some of the text  is written in the present tense, some in the present perfect and some in the past. Please correct it, unify, and then have it professionally proofread.]

Response 4: Agree. Thank you for your reminder.

I have changed 'perform' to the past tense 'performed' and have checked and modified other similar issues in the material method.

Comments 5: [line 128 :4where? it was in the fridge? walkingcooler? phytothrone? growth chamber?]

Response 5: Agree. Thank you for your reminder.

In line 134, the description of temperature environment has been added.

Comments 6: [line 135 :what was light intensity?.]

Response 6: Agree. Thank you for your reminder.

The light intensity was set to 1000 lux. It had beed added in line 137.

Comments 7: [line138-139 Figure 1 : Captions of Tables and Figures have to be precise and detailed, so that reader can understand what they represent without looking at the text of publication.]

Response 7: Agree. Thank you for your reminder.

At the end of the title in Table 1,' of early-maturing Pyrus perifolia seeds ' has been added. All the seeds in the four methods were placed at a cold temperature of 4 for 60 days. The changes can be found in figure 1(Line 140-142)

Comments 8: [line 155 : provide references to all and producer of each.]

Response 8: Agree. Thank you for your reminder.

Relevant references had been added in this section. 7.

Comments 9: [line 199 : what you mean by "conventional method"? stratification in sand? if so please decribe this method as conventional also in Mat&Met chapter]

Response 9: Agree. Thank you for your reminder.

This part of the description was not accurate, and ‘the controversial’ had been deleted in line 203.

Comments 10: [line 206 : cold temperature temperature, please add the info about the exact temperature value and its duration; with >during in vitro]

Response 10: Agree. Thank you for your reminder.

The temperature and time of treatment are added in line 209.

‘with culture medium germination’ had been changed to ‘during in vitro’  according to your suggestion.

Comments 11: [line 245 : change MS culture to : In vitro culture.]

Response 11: Agree. Thank you for your reminder.

MS culture  had been changed to : In vitro culture.The change can be found in line 249.

Comments 12: [line 273-274 : please correct the style of this sentence.]

Response 12: Agree. Thank you for your reminder.

The description of this part had been simplified and modified to ensure the fluency and accuracy of the language. In line 277-278

Comments 13: [line 277:ideal?.]

Response 13: Agree. Thank you for your reminder.

Ideal had been changed to best according your suggestion.

Comments 14: [[line 284:supplement with a detailed description of the treatments and media and explain the symbols.]

Response 14: Agree. Thank you for your reminder.

‘Five different mediums media( MS, C167, White , G398 and M519) were selected, all the seeds were incubated at 4°C for 100 days’ had beed add to descripte the details.

Comments 15: [line 422: seed >complete seed coat removal.]

Response 15: Agree. Thank you for your reminder.

This part of the description was not accurate, and ‘complete seed coat removal’ had been changed in line 427.

4. Response to Comments on the Quality of English Language

Point 1: line 24 ‘could increase’ : increased (the past tense)

Response 1:    The ‘could increase’ had beed changed to’increased’. In line 24

Point 2: line 31 : line 25 ‘could improve’ : improved (the past tense)

Response 2:   ‘could improve’ change to ‘improved’. In line25

Point 3: line 55 : In vitro ~~ --> Not Italic

Response 3:    ‘In vitro ‘ changed to ‘In vitro’. In line 56.. This type of change has already been applied to the entire manuscript

Point 4: line 143 : in? or on. please apply the correct version to all text

Response 4:    It is more appropriate to use "on", and it has also been revised in other contents of the manuscript.

Point 5: line 158 : 150m --> 150ml

Response 5:   Incomplete input, which has been modified to ‘150ml’.

Point 6: line 177 : above or below? Because the calculations are described in Statistical analysis chapter

Response 6:   change ‘above ‘to ‘below’ in line 180

Point 7: line 229 : suggests or demonstrates?

Response 7:   change ‘suggests ‘to ‘demonstrates’ in line 233

Point 8: line 423 : and --> on

Response 8:   Adjust and in line 428 to on
